# MULTIPLE OUTPUT SAMPLES FOR EACH INPUT IN A SINGLE-OUTPUT GAUSSIAN PROCESS

## ABSTRACT

The standard Gaussian Process (GP) is formulated to only consider a single output sample for each input in the training set. Datasets for subjective tasks, such as spoken language assessment, may be annotated with output labels from multiple human raters for each input. This paper proposes to generalise the GP to allow for multiple output samples per input in the training set. This differs from a multi-output GP, because all output samples are from the same task here. The output density function is formulated to be the joint likelihood of observing all output samples. Through this, the hyper-parameters are optimised using a criterion that is similar to minimising a Kullback-Leibler divergence. This is computationally cheaper than repeating the input for each output sample. The test set predictions are inferred fairly similarly to a standard GP, with a key difference being in the optimised hyper-parameters. This approach is evaluated on spoken language assessment tasks, using the public speechocean762 dataset and an internal Tamil language dataset. The results show that by using the proposed method, the GP is able to compute a test set output distribution that is more similar to the collection of reference outputs annotated by multiple human raters.

## 1 INTRODUCTION

The Gaussian Process (GP) (Rasmussen & Williams, 2006) expresses a prediction uncertainty that naturally increases for inputs further away from the training data. As opposed to this, Neural Networks (NN) have been observed to yield overly confident predictions, even when the input is from a mismatched domain (Guo et al., 2017). This behaviour of a GP may allow better explainability of the model's predictions. Having explainable predictions of uncertainty may be especially desirable for tasks that are subjective in nature. In such subjective tasks, multiple human annotators may provide differing output labels for the same input. A collection of human annotations for the same input may therefore be interpreted as a reference of uncertainty that an automatic model should also aim to compute. In such settings, the uncertainties expressed by the model and the human annotators can be explicitly compared.

However, the standard GP formulation assumes that each input in the training set is paired with only a single output, which is treated as the ground truth. This paper proposes to extend the GP formulation, to accommodate for situations where multiple samples of output labels for the same task are provided for each input. The hyper-parameters can be optimised and the test set predictions can be inferred, with the consideration of having multiple training set output samples, in a computationally cheaper manner than simply repeating the inputs for each output sample.

## 2 RELATED WORK

The multi-output GP is formulated in a multi-task framework (Yu et al., 2005; Bonilla et al., 2007). This treats the multiple outputs for each input as separate tasks. On the other hand, this paper considers a single-output GP with a single output task, where multiple output samples for each input are present for that task.

This paper considers optimising the GP hyper-parameters using a criterion that is similar to minimising a distance to a reference output density function. When training a NN, the reference output

can be in the form of a distribution, as opposed to a scalar or single class. Full-sum training in speech recognition (Yan et al., 1997) and handwriting recognition (Senior & Robinson, 1995) trains a NN toward the distributional reference formed by the soft forced alignment. In BLIND, NNs for Spoken Language Assessment (SLA) are trained toward the distribution represented by the scores from multiple human raters. The distributional output from one NN can also be used as a reference to train another NN toward (Li et al., 2014; Hinton et al., 2014).

## 3 GAUSSIAN PROCESS REGRESSION

When given a collection of $N$ input feature vectors of dimension $D$, $\boldsymbol{X} \in \mathbb{R}^{N \times D}$, a GP places a jointly Gaussian prior over latent variables, $\boldsymbol{f} \in \mathbb{R}^N$, as

$$p\left(\boldsymbol{f}|\boldsymbol{X}\right) = \mathcal{N}\left(\boldsymbol{f}; \boldsymbol{0}, \boldsymbol{K}\left(\boldsymbol{X}, \boldsymbol{X}\right)\right). \tag{1}$$

Here, $p(\boldsymbol{f}|\boldsymbol{X})$ is an abbreviation of $p(\mathring{\boldsymbol{f}} = \boldsymbol{f}|\mathring{\boldsymbol{X}} = \boldsymbol{X})$, with the interpretation of being the likelihood of continuous random variables $\mathring{\boldsymbol{f}}$ taking the values $\boldsymbol{f}$. A multivariate Gaussian density function with mean $\boldsymbol{\mu}$ and covariance $\boldsymbol{V}$ is represented as $\mathcal{N}(\boldsymbol{f}; \boldsymbol{\mu}, \boldsymbol{V})$. In a GP, the covariance of the latent variable is defined as a pair-wise distance between the inputs, with the notion of distance being defined by the kernel, $\boldsymbol{K}$. In this paper, the squared exponential kernel is used, with kernel matrix elements defined as

$$k_{ij}\left(\boldsymbol{X}, \boldsymbol{X}\right) = s^2 \exp\left[-\frac{\left(\boldsymbol{x}_i - \boldsymbol{x}_j\right)^\top \left(\boldsymbol{x}_i - \boldsymbol{x}_j\right)}{2l^2}\right], \tag{2}$$

where $i$ and $j$ are the matrix indexes, $l$ is a length hyper-parameter, and $s$ is a scale hyper-parameter. The GP makes the assumption that the outputs, $\boldsymbol{y} \in \mathbb{R}^N$, are conditionally independent of the inputs when given the latent variables. The outputs are Gaussian distributed about a mean of the latent variable, with a noise hyper-parameter, $\sigma$, to accommodate for observational noise,

$$p\left(\boldsymbol{y}|\boldsymbol{f}\right) = \mathcal{N}\left(\boldsymbol{y}; \boldsymbol{f}, \sigma^2 \boldsymbol{I}\right), \tag{3}$$

where $\boldsymbol{I}$ is the identity matrix.

### 3.1 TRAINING

Training a GP involves estimating the hyper-parameters of the kernel, $s$ and $l$, and of the observation noise $\sigma$. One approach is to find the hyper-parameters that maximise the marginal log-likelihood of the training data,

$$\mathcal{F} = \log p\left(\boldsymbol{y}^{\text{ref}}|\boldsymbol{X}\right), \tag{4}$$

where the training data comprises pairs of observed features and reference outputs, $\boldsymbol{y}^{\text{ref}}$. The marginal likelihood can be computed as

$$p\left(\boldsymbol{y}|\boldsymbol{X}\right) = \int p\left(\boldsymbol{y}|\boldsymbol{f}\right) p\left(\boldsymbol{f}|\boldsymbol{X}\right) d\boldsymbol{f} \tag{5}$$

$$= \mathcal{N}\left(\boldsymbol{y}; \boldsymbol{0}, K\left(\boldsymbol{X}, \boldsymbol{X}\right) + \sigma^2 \boldsymbol{I}\right). \tag{6}$$

Optimising the hyper-parameters using gradient-based methods requires the inversion of $K(\boldsymbol{X}, \boldsymbol{X}) + \sigma^2 \boldsymbol{I}$ (Rasmussen & Williams, 2006), which entails a number of computational operations that scale as $\mathcal{O}(N^3)$, when using Gaussian elimination or when computing the singular value decomposition in a pseudo-inverse implementation. Algorithms, such as Petković & Stanimirović (2009), are able to reduce the polynomial power, but still require more than $\mathcal{O}(N^2)$ operations.

### 3.2 INFERENCE

When performing evaluation, a test set of input feature vectors, $\widehat{\boldsymbol{X}}$, is given, and the task is to infer the predicted outputs, $\widehat{\boldsymbol{y}}$. Inference through a GP can be performed by first computing the density

function over the predicted latent variables,

$$p\left(\widehat{\boldsymbol{f}}\Big|\widehat{\boldsymbol{X}}, \boldsymbol{y}, \boldsymbol{X}\right) = \frac{p\left(\boldsymbol{y}, \widehat{\boldsymbol{f}}\Big|\boldsymbol{X}, \widehat{\boldsymbol{X}}\right)}{p\left(\boldsymbol{y}|\boldsymbol{X}\right)} \tag{7}$$

$$= \frac{\mathcal{N}\left(\begin{bmatrix}\boldsymbol{y}\\ \widehat{\boldsymbol{f}}\end{bmatrix}; \boldsymbol{0}, \begin{bmatrix}\boldsymbol{K}\left(\boldsymbol{X}, \boldsymbol{X}\right) + \sigma^2\boldsymbol{I} & \boldsymbol{K}\left(\boldsymbol{X}, \widehat{\boldsymbol{X}}\right)\\ \boldsymbol{K}\left(\widehat{\boldsymbol{X}}, \boldsymbol{X}\right) & \boldsymbol{K}\left(\widehat{\boldsymbol{X}}, \widehat{\boldsymbol{X}}\right)\end{bmatrix}\right)}{\mathcal{N}\left(\boldsymbol{y}; \boldsymbol{0}, K\left(\boldsymbol{X}, \boldsymbol{X}\right) + \sigma^2\boldsymbol{I}\right)} \tag{8}$$

$$= \mathcal{N}\left(\widehat{\boldsymbol{f}}; \widehat{\boldsymbol{\mu}}, \widehat{\boldsymbol{V}}\right), \tag{9}$$

where

$$\widehat{\boldsymbol{\mu}} = \boldsymbol{K}\left(\widehat{\boldsymbol{X}}, \boldsymbol{X}\right)\left[\boldsymbol{K}\left(\boldsymbol{X}, \boldsymbol{X}\right) + \sigma^2\boldsymbol{I}\right]^{-1}\boldsymbol{y} \tag{10}$$

$$\widehat{\boldsymbol{V}} = \boldsymbol{K}\left(\widehat{\boldsymbol{X}}, \widehat{\boldsymbol{X}}\right) - \boldsymbol{K}\left(\widehat{\boldsymbol{X}}, \boldsymbol{X}\right)\left[\boldsymbol{K}\left(\boldsymbol{X}, \boldsymbol{X}\right) + \sigma^2\boldsymbol{I}\right]^{-1}\boldsymbol{K}\left(\boldsymbol{X}, \widehat{\boldsymbol{X}}\right), \tag{11}$$

which, similarly to training, also requires the computation of $[\boldsymbol{K}(\boldsymbol{X}, \boldsymbol{X}) + \sigma^2\boldsymbol{I}]^{-1}$.

The predictive density function for the output can then be computed as

$$p\left(\widehat{\boldsymbol{y}}\Big|\widehat{\boldsymbol{X}}, \boldsymbol{y}, \boldsymbol{X}\right) = \mathcal{N}\left(\widehat{\boldsymbol{y}}; \widehat{\boldsymbol{\mu}}, \widehat{\boldsymbol{V}} + \sigma^2\boldsymbol{I}\right). \tag{12}$$

Finally, the predicted scalar output for each test data point, $i$, can be inferred from equation 12, using a decision rule. One possible decision rule is to choose the mean of equation 12, $\widetilde{y}_i^\star = \widehat{\mu}_i$. As a result of the symmetry and uni-modality of a Gaussian density function, the decision rules of choosing either the mean, mode, or median of equation 12 are equivalent.

## 4  MULTIPLE OUTPUT SAMPLES

In the standard GP formulation, each input data point is paired with a single output sample. As is described in section 5, situations may arise where modelling multiple output samples for each input data point may be beneficial. This paper considers such a scenario, whereby all output samples for the same input data point are drawn from the same output density $p(\boldsymbol{y}|\boldsymbol{f})$ and kernel $\boldsymbol{K}$. It is thus assumed that these multiple output samples belong to the same task. This differs from the multi-output GP formulation, which assumes that each separate output represents a different task.

Let $i$ be the training set data point index. Let each input training data point, $\boldsymbol{x}_i$, be associated with $R$ output reference samples, $\boldsymbol{y}_i^{\text{ref}} = [y_{i1}^{\text{ref}}, \cdots, y_{iR}^{\text{ref}}]$, where $y_{ir}^{\text{ref}}$ is the $r$th reference output sample for the $i$th training data point. It is assumed here for simplicity that the number of output samples is the same for all inputs, but an analogous formulation can be derived to allow for varying numbers of output samples. These outputs are stacked together across all input data points to yield an output matrix $\boldsymbol{Y}^{\text{ref}} \in \mathbb{R}^{N \times R}$.

### 4.1  REPETITION OF INPUTS

A naive approach to accommodate multiple output samples may be to simply flatten $\boldsymbol{Y}^{\text{ref}}$ into a vector and repeat the inputs, $\boldsymbol{x}_i$, for each output sample. This will yield a $\mathbb{R}^{NR \times NR}$ kernel of $\widetilde{\boldsymbol{K}} = \boldsymbol{K}(\boldsymbol{X}, \boldsymbol{X}) \otimes \boldsymbol{1}^{(R)1}$, where $\boldsymbol{1}^{(R)}$ is a $\mathbb{R}^{R \times R}$ matrix of 1s and $\otimes$ represents a Kronecker (tensor) product. Although it may be theoretically questionable as to what it may mean to have such a non-full-rank kernel to represent the Gaussian covariance in the GP, there is no need to explicitly compute its inverse. Instead, only the inverse of $\widetilde{\boldsymbol{K}} + \sigma^2\boldsymbol{I}$ needs to be computed during training and inference, which is full-rank. Computing this matrix inverse requires a number of operations that scales as $\mathcal{O}(N^3R^3)$, as opposed to $\mathcal{O}(N^3)$ when using only a single output sample. Computational constraints may then impose the need for sparse approximation methods, such as Quiñonero-Candela & Rasmussen (2005), even for modest training set sizes.

---

[1] A $\sim$ above a matrix or vector is used here to emphasise that it comprises repeated elements.

## 4.2 Omission of Redundancy in the Prior

When using this repeated kernel, the latent variables are distributed as $p(\widetilde{\boldsymbol{f}}|\boldsymbol{X}) = \mathcal{N}(\widetilde{\boldsymbol{f}}|\boldsymbol{0}, \widetilde{\boldsymbol{K}})$. The repeated structure of $\widetilde{\boldsymbol{K}}$ causes all latent variables that are associated with the same training data point to be perfectly correlated. Therefore, it seems computationally redundant to separately model the repeated latent variables. Instead, this paper proposes that only a single instance of each latent variable per training data point be modelled using equation 1, and the joint density of the multiple output samples be conditioned on these non-repeated latent variables, $p(\boldsymbol{Y}|\boldsymbol{f})$. The diagonal covariance in the output density function of a standard GP in equation 3 implies that the outputs from different training data points are conditionally independent of each other when given the latent variables. This conditional independence, applied to the case of repeated inputs in the naive approach, yields in this proposed approach a joint output density that can be factorised as

$$p\left(\boldsymbol{Y}|\boldsymbol{f}\right) = \prod_{r=1}^{R} \mathcal{N}\left(\boldsymbol{y}_r; \boldsymbol{f}, \sigma^2 \boldsymbol{I}\right) \tag{13}$$

$$= \mathcal{N}\left(\overline{\boldsymbol{\mu}}\left(\boldsymbol{Y}\right); \boldsymbol{f}, \frac{\sigma^2}{R}\boldsymbol{I}\right) \mathcal{N}\left(\overline{\boldsymbol{\eta}}\left(\boldsymbol{Y}\right); \boldsymbol{0}, \frac{\sigma^2}{R}\boldsymbol{I}\right) \left(2\pi\sigma^2\right)^{(2-R)\frac{N}{2}} R^{-N}, \tag{14}$$

where

$$\overline{\boldsymbol{\mu}}\left(\boldsymbol{Y}\right) = \frac{1}{R}\sum_{r=1}^{R} \boldsymbol{y}_r \quad \text{and} \quad \overline{\eta}_i\left(\boldsymbol{y}_i\right) = \sqrt{\frac{1}{R}\sum_{r=1}^{R} \left(y_{ir} - \overline{\mu}_i\left(\boldsymbol{y}_i\right)\right)^2}, \tag{15}$$

and $\boldsymbol{y}_r \in \mathbb{R}^N$ is a vector of the $r$th output sample for all inputs. The factorisation and diagonal covariance in equation 13 assume that the multiple output samples for the same input and across different inputs are independent of each other, when given the latent variable. As a result, the output density does not depend on the order of the multiple output samples for each input. This can be explicitly seen in the re-parameterisation of the multiple outputs as the empirical mean $\overline{\boldsymbol{\mu}}$ and empirical biased standard deviation $\overline{\boldsymbol{\eta}}$, both of which are independent of the ordering of the output samples. Also, equation 14 may seem unnormalised over $\overline{\boldsymbol{\mu}}$ and $\overline{\boldsymbol{\eta}}$. However, it should be noted that $p(\boldsymbol{Y}|\boldsymbol{f})$ is expected to sum to one when integrated over $\boldsymbol{Y}$, and not over $\overline{\boldsymbol{\mu}}$ and $\overline{\boldsymbol{\eta}}$.

### 4.2.1 Training

The marginal likelihood can be computed from the non-redundant prior of equation 1 and the output density of equation 14, as

$$p\left(\boldsymbol{Y}|\boldsymbol{X}\right) = \int p\left(\boldsymbol{Y}|\boldsymbol{f}\right) p\left(\boldsymbol{f}|\boldsymbol{X}\right) d\boldsymbol{f} \tag{16}$$

$$= \mathcal{N}\left(\overline{\boldsymbol{\mu}}\left(\boldsymbol{Y}\right); \boldsymbol{0}, \boldsymbol{K}\left(\boldsymbol{X}, \boldsymbol{X}\right) + \frac{\sigma^2}{R}\boldsymbol{I}\right) g\left(\boldsymbol{Y}\right), \tag{17}$$

where

$$g\left(\boldsymbol{Y}\right) = \mathcal{N}\left(\overline{\boldsymbol{\eta}}\left(\boldsymbol{Y}\right); \boldsymbol{0}, \frac{\sigma^2}{R}\boldsymbol{I}\right) \left(2\pi\sigma^2\right)^{(2-R)\frac{N}{2}} R^{-N}. \tag{18}$$

Analogously to the standard GP, and equivalently to the naive approach, the hyper-parameters of the proposed approach can be optimised by maximising the marginal log-likelihood of all output samples,

$$\mathcal{F} = \log p\left(\boldsymbol{Y}^{\text{ref}}|\boldsymbol{X}\right). \tag{19}$$

Unlike a naive implementation, gradient-based optimisation of the proposed method has a number of operations that scales as $\mathcal{O}(N^3)$, similarly to a standard GP with a single output sample.

In appendix A, maximising equation 19 is shown to be related to minimising a Kullback-Leibler (KL) divergence between the marginal likelihood and the reference density function over output samples, under several assumptions. This suggests that maximising equation 19 may encourage the GP's output density to be more similar to the reference density.

### 4.2.2 INFERENCE

When performing evaluation, the predicted output for each input test data point can be inferred from the predictive density function. To compute the predictive density function, the joint density between the training set outputs and the test latent variables first needs to be computed. When the training set comprises multiple output samples per input, this paper again proposes that there is no need to separately model redundant latent variables, by expressing this joint density as

$$p\left(\boldsymbol{Y}, \widehat{\boldsymbol{f}} \middle| \boldsymbol{X}, \widehat{\boldsymbol{X}}\right) = \int p\left(\boldsymbol{f}, \widehat{\boldsymbol{f}} \middle| \boldsymbol{X}, \widehat{\boldsymbol{X}}\right) p\left(\boldsymbol{Y} | \boldsymbol{f}\right) d\boldsymbol{f} \tag{20}$$

$$= \int \mathcal{N}\left(\begin{bmatrix} \boldsymbol{f} \\ \widehat{\boldsymbol{f}} \end{bmatrix}; \boldsymbol{0}, \begin{bmatrix} \boldsymbol{K}\left(\boldsymbol{X}, \boldsymbol{X}\right) & \boldsymbol{K}\left(\boldsymbol{X}, \widehat{\boldsymbol{X}}\right) \\ \boldsymbol{K}\left(\widehat{\boldsymbol{X}}, \boldsymbol{X}\right) & \boldsymbol{K}\left(\widehat{\boldsymbol{X}}, \widehat{\boldsymbol{X}}\right) \end{bmatrix}\right) \mathcal{N}\left(\overline{\boldsymbol{\mu}}\left(\boldsymbol{Y}\right); \boldsymbol{f}, \frac{\sigma^2}{R}\boldsymbol{I}\right) g\left(\boldsymbol{Y}\right) d\boldsymbol{f} \tag{21}$$

$$= \mathcal{N}\left(\begin{bmatrix} \overline{\boldsymbol{\mu}}\left(\boldsymbol{Y}\right) \\ \widehat{\boldsymbol{f}} \end{bmatrix}; \boldsymbol{0}, \begin{bmatrix} \boldsymbol{K}\left(\boldsymbol{X}, \boldsymbol{X}\right) + \frac{\sigma^2}{R}\boldsymbol{I} & \boldsymbol{K}\left(\boldsymbol{X}, \widehat{\boldsymbol{X}}\right) \\ \boldsymbol{K}\left(\widehat{\boldsymbol{X}}, \boldsymbol{X}\right) & \boldsymbol{K}\left(\widehat{\boldsymbol{X}}, \widehat{\boldsymbol{X}}\right) \end{bmatrix}\right) g\left(\boldsymbol{Y}\right). \tag{22}$$

Analogously to equation 7, the predictive density over the test latent variables is

$$p\left(\widehat{\boldsymbol{f}} \middle| \widehat{\boldsymbol{X}}, \boldsymbol{Y}, \boldsymbol{X}\right) = \frac{p\left(\boldsymbol{Y}, \widehat{\boldsymbol{f}} \middle| \boldsymbol{X}, \widehat{\boldsymbol{X}}\right)}{p\left(\boldsymbol{Y} | \boldsymbol{X}\right)} \tag{23}$$

$$= \mathcal{N}\left(\widehat{\boldsymbol{f}}; \breve{\boldsymbol{\mu}}, \breve{\boldsymbol{V}}\right), \tag{24}$$

where

$$\breve{\boldsymbol{\mu}} = \boldsymbol{K}\left(\widehat{\boldsymbol{X}}, \boldsymbol{X}\right)\left[\boldsymbol{K}\left(\boldsymbol{X}, \boldsymbol{X}\right) + \frac{\sigma^2}{R}\boldsymbol{I}\right]^{-1}\overline{\boldsymbol{\mu}}\left(\boldsymbol{Y}\right) \tag{25}$$

$$\breve{\boldsymbol{V}} = \boldsymbol{K}\left(\widehat{\boldsymbol{X}}, \widehat{\boldsymbol{X}}\right) - \boldsymbol{K}\left(\widehat{\boldsymbol{X}}, \boldsymbol{X}\right)\left[\boldsymbol{K}\left(\boldsymbol{X}, \boldsymbol{X}\right) + \frac{\sigma^2}{R}\boldsymbol{I}\right]^{-1}\boldsymbol{K}\left(\boldsymbol{X}, \widehat{\boldsymbol{X}}\right). \tag{26}$$

The predictive density over the test outputs is then

$$p\left(\widehat{\boldsymbol{y}} \middle| \widehat{\boldsymbol{X}}, \boldsymbol{Y}, \boldsymbol{X}\right) = \mathcal{N}\left(\widehat{\boldsymbol{y}}; \breve{\boldsymbol{\mu}}, \breve{\boldsymbol{V}} + \sigma^2\boldsymbol{I}\right). \tag{27}$$

Finally, the output can be inferred for each test input data point by choosing the mean of this predictive density, similarly to how a score is chosen from equation 12.

In this proposed approach, the matrix inverted in equations 25 and 26 is of size $\mathbb{R}^{N \times N}$, as opposed to $\mathbb{R}^{NR \times NR}$ in the naive repetition approach. The number of operations thus scales as $\mathcal{O}(N^3)$, similarly to a standard GP with a single output sample.

The covariance of the predictive densities in both equations 12 and 27 are independent of the training set outputs. These covariances depend only on the training set inputs and the hyper-parameters. As such, information about the proximity between the multiple output samples for the same input in the training set does not influence the resulting predictive covariance. In fact, the GP formulation is such that whether the training set output values are near or far from each other, for two inputs that are near as measured by the kernel, does not influence the resulting predictive covariance for test inputs near to these training inputs. What does affect the predictive covariance is the distance between the test and training set inputs. This implies that a GP may be more suited for computing distributional uncertainty, and less so for data uncertainty (Xu et al., 2021). Perhaps, a future research direction may be to allow the GP to take training set output information into account for the predictive covariance.

## 5 SPOKEN LANGUAGE ASSESSMENT

Section 4 proposes a modification of a single-output GP, to allow the training set to comprise multiple output samples for each input, where it is assumed that all output samples for the same input are

related to the same task. This formulation may be applicable for tasks that are not strictly deterministic. For example, noise in the environment may induce a variance in the observed output, a task may have multiple valid output values that can correspond to the same input value, or a task may be subjective such that different human experts may annotate the same input with different output values. In such scenarios, including multiple output samples per input data point in the training set may be more informative than using only a single output sample for each input.

## 5.1 TASK

Examples of subjective tasks are language assessment, emotion recognition (Busso et al., 2008), and medical diagnostics. The experiments in this paper are applied to SLA. In SLA, the input is an audio recording of speech from a student. The task is to predict a score for that input that is similar to a score that an expert human rater would have given. This score is meant to relate to aspects of the oral proficiency of the student. Examples of potential assessed aspects include the pronunciation accuracy, intonation, fluency, and prosody. In practice, examination committees often try to standardise the assessment criteria through a rubric. However, it may be difficult to concisely document a rubric that takes all possible circumstances into account. As such, much may still be left to the human rater's subjective opinion when assigning a score.

To account for this subjectivity, training datasets for SLA may include the reference scores from multiple human raters for each input. It is standard practice to compute a combined reference score, by taking either the majority vote (Lin & Wang, 2021), median (Zhang et al., 2021b), or mean (Zhang et al., 2021b) of these multiple scores. An automatic SLA model may then be trained toward and evaluated against this combined reference. Examples of such approaches are Metallinou & Cheng (2014); Duan & Chen (2020); Lin & Wang (2021) using NNs, and van Dalen et al. (2015) using GPs. However, collapsing the scores from multiple human raters into a single score may result in a loss of information about the output uncertainty that should be associated with each input. The proposed framework allows a GP to take into account multiple output samples for each input in the training set.

As can be seen from equation 25, the predictive mean under the proposed formulation is dependent on the mean of the multiple output samples, $\overline{\mu}$, and independent of $\overline{\eta}$. This is similar to a standard GP formulation, when using a mean combination of the multiple rater scores to get the reference. As mentioned previously, the predictive covariance is independent of the training set outputs. Thus, during inference, the remaining difference between the standard GP and the proposed modification is in the hyper-parameters. The proposed approach optimises the hyper-parameters by maximising the marginal log-likelihood of all training set outputs, which may be related to minimising the KL divergence between this marginal likelihood and the reference density function. As opposed to this, the standard approach maximises the marginal log-likelihood of the combined output reference.

## 5.2 EVALUATION MEASURES

Previous works on SLA often evaluate a model's performance by comparing the predicted output score from the model against the combined reference score. Two evaluation measures that are often used are Pearson's Correlation Coefficient (PCC) and Mean Squared Error (MSE) (Zhang et al., 2021b). These are written out in appendix B. Both measure how well the predicted score matches with the combined reference score. This assumes that the combined reference score is correct.

However, comparing only against the combined reference score omits information about the agreement between the multiple human raters. This agreement may be representative of a reference of the uncertainty that should be associated with each input. It may be useful to assess how well the predictive density agrees with this reference uncertainty. This is measured using the KL divergence between the predictive and reference densities, through both the unnormalised continuous and the discrete forms described in appendix C.

## 6 EXPERIMENTS

### 6.1 DATASETS

Experiments were performed on the speechocean762 dataset (Zhang et al., 2021b) and an internal Tamil language dataset. These are regression tasks with multiple output reference samples provided. The public speechocean762 dataset allows for experimental repeatability, while the internal Tamil dataset is used to verify that the observed trends are generalisable. Speechocean762 comprises a training and test set, each with 2500 sentences and 125 disjoint speakers. The native Mandarin speakers read the sentences in English. Each sentence is annotated with a variety of score types at different linguistic levels, but only the sentence-level pronunciation accuracy was used in this paper. Each sentence is annotated by 5 human raters, each assigning an integer score between 0 and 10.

The Tamil language dataset (Zhang et al., 2021a) comprises a total of 4211 sentences, read by 100 speakers, aged between 9 to 16 years, from Singapore. This was split into training and test sets with 2118 and 2093 sentences respectively, with speaker disjointment. Stratified sampling was used to split the data, to ensure similar demographics between the different educational levels and assessment scores across the two sets. Each sentence was annotated with pronunciation accuracies by 4 human raters, with integer scores between 1 and 5.

For both datasets, combined reference scores were computed as a mean of the multiple rater scores. This differs from Zhang et al. (2021b), which instead computes the median score. Using the mean may yield a baseline that is a more appropriate match for the formulation in equation 14. Following the standard in Zhang et al. (2021b), the predicted and combined reference scores were first rounded to the closest integers, before computing the PCC and MSE, for both datasets.

These datasets are chosen to assess the proposed approach for three reasons. First, the training set sizes allow for computational feasibility with a GP, without having to resort to sparse approximations or to use only a subset of the training data. The use of sparse approximations may potentially occlude experimental trends. Second, both datasets are annotated with multiple output reference samples for each training data point, thereby allowing the specific experiments in this paper to be run. Third, these are regression tasks, thereby avoiding any complications that may arise from using a GP for classification. Datasets for emotion recognition and stutter detection are also often annotated with multiple output reference samples. Potential emotion recognition datasets that can be considered for future exploration include IEMOCAP (Busso et al., 2008) with a regression task annotated with two output samples and may thus be too few to observe interpretable trends, and MSP-Podcast (Lotfian & Busso, 2019) with a regression task annotated with five output samples but has a much larger training set. For stutter detection, SEP-28k (Lea et al., 2021) and KSoF (Bayerl et al., 2022) are classification tasks annotated with three output samples.

### 6.2 MODELS

The feature extraction for a NN SLA model was similar to Zhang et al. (2021a). First, speech recognition models described in appendix D were trained. These were then used to force align the audio with the transcriptions. From the forced alignment, Goodness Of Pronunciation (GOP) (Witt & Young, 2000), Log Phone Posterior (LPP) (Hu et al., 2015), Log Posterior Ratio (LPR) (Hu et al., 2015), and tempo (Zhang et al., 2021a) features were extracted for each phone. Pitch features (Ghahremani et al., 2014) were extracted from the audio, and pooled across all frames within each phone. A continuous skip-gram model (Mikolov et al., 2013), with one recurrent NN (Elman, 1990) hidden layer having 32 nodes, was trained on the training set non-silence phone sequences, and used to extract phone embeddings (Zhang et al., 2021a). The GOP, LPP, LPR, tempo, pitch, and phone embedding features were concatenated to form one feature vector per phone.

In baseline NN SLA models, the sequence of feature vectors, with length equal to the number of phones in the sentence, were fed into a Bidirectional Long Short-Term Memory (BLSTM) layer (Graves & Schmidhuber, 2005), with 32 nodes per direction. The output was then pooled across all phones in the sequence with equal weights, and fed through a linear layer. Two forms of NN output layers were used. In the first variation, referred to as $NN_{scalar}$, the linear layer had a single dimensional output, which then passed through a sigmoid (Zhang et al., 2021a). This sigmoid output was then scaled to the bounds of the output score range, and the computed output was treated as the predicted score. Such a regression model was trained by minimising the MSE toward the combined

reference score. In the second variation, referred to as $NN_{categorical}$, the linear layer output dimension was equal to the number of integer scores, and the output was fed through a softmax (Duan & Chen, 2020). This classification model was trained using the cross-entropy criterion toward the combined reference score. Both models used dropout (Srivastava et al., 2014) with an omission probability of 60%, before the BLSTM and linear layers.

It may not be straight forward to allow a GP to utilise sequential inputs. Work in Lodhi et al. (2000) investigates designing kernels to operate on sequences, while van Dalen et al. (2015) extracts hand-crafted sentence-level features for SLA. In this paper, the activations after the NN pooling layer were treated as bottleneck features (Konig et al., 1998), and used as sentence-level inputs to the GP. A principle component analysis whitening transformation, estimated on the training set, was applied to these features to better abide by the diagonal covariance assumption of the scalar length hyper-parameter in the kernel in equation 2

## 6.3 STATISTICAL SIGNIFICANCE

The statistical significance, $\rho$, of the difference between either the MSE or KL divergence evaluation measures of two models was computed using a two-tailed paired $t$-test. As seen in equation 33, the PCC is not easily expressed as a sum over data points, making it difficult to apply the central limit theorem. The significance was instead computed using the $Z_1^\star$ approach in (Steiger, 1980). Here, an approximately normally distributed transformation (Dunn & Clark, 1969) was first computed from the two PCCs being compared. Then the significance was computed as the two-tailed cumulative density of this transformed quantity.

## 6.4 BOTTLENECK FEATURES AS INPUTS TO A GAUSSIAN PROCESS

The first experiment investigates the soundness of using bottleneck features as inputs to a GP, and the impact that the training approach used for the NN bottleneck feature extractor can have. The $NN_{scalar}$ and $NN_{categorical}$ models were trained as described in section 6.2. Sentence-level bottleneck features from after the pooling layer were computed, and used as inputs to the GP. The standard GP was used, with only a single reference output sample per training data point, referred to as $GP_{base}$. The GP hyper-parameters were optimised separately for each of the feature types, by using gradient descent to maximise the marginal log-likelihood of the training set, as in equation 4. The mean combined score between the multiple raters was used as the training set reference output.

Table 1: Impact of the bottleneck feature extractor on a Gaussian process

| Dataset | SLA model | Bottleneck extractor | PCC | MSE |
|---|---|---|---|---|
| speechocean762 | $NN_{scalar}$ | - | 0.711 | 1.232 |
| | $NN_{categorical}$ | - | 0.701 | 1.208 |
| | $GP_{base}$ | $NN_{scalar}$ | 0.710 | 1.149 |
| | $GP_{base}$ | $NN_{categorical}$ | 0.694 | 1.209 |
| Tamil | $NN_{scalar}$ | - | 0.638 | 0.716 |
| | $NN_{categorical}$ | - | 0.599 | 0.621 |
| | $GP_{base}$ | $NN_{scalar}$ | 0.565 | 0.641 |
| | $GP_{base}$ | $NN_{categorical}$ | 0.555 | 0.648 |

The results in table 1 show that both the $NN_{scalar}$ and $NN_{categorical}$ models yield comparable PCC and MSE performances, with no clear victor over both measures. A GP that uses the $NN_{scalar}$ bottleneck features may yield better PCC and MSE performances than when using $NN_{categorical}$ features. The statistical significance between GPs that use either type of features are $\rho_{PCC} = 0.033$ and $\rho_{MSE} = 0.078$ for speechocean762, and $\rho_{PCC} = 0.334$ and $\rho_{MSE} = 0.617$ for Tamil. Thus the performance differences may not be significant. Although not significant, the improvements from using bottleneck features from $NN_{scalar}$ are consistent across both datasets. Therefore, all subsequent experiments use bottleneck features from $NN_{scalar}$ as inputs to the GP. The GP and NN SLA models exhibit comparable performances.

### 6.5 MULTIPLE OUTPUT SAMPLES PER INPUT DATA POINT

The next experiment assesses the proposed extension, of allowing the GP to use the separate output scores from multiple human raters in the training set, referred to as $GP_{joint}$. This is compared against $GP_{base}$, and also against a naive multiple output reference samples extension of repeating the training set input features, referred to as $GP_{repeat}$. The $GP_{joint}$ hyper-parameters were optimised by maximising equation 19, and the predicted scores were inferred using equation 27.

Table 2: Using training set scores from multiple raters in a Gaussian process

| Dataset | Model | PCC | MSE | KL divergence continuous | discrete | Inference time (s) |
|---|---|---|---|---|---|---|
| speechocean762 | $GP_{base}$ | 0.710 | 1.149 | 5.04 | 3.10 | 136 ±3 |
| | $GP_{repeat}$ | 0.713 | 1.136 | 1.88 | 0.85 | 8476 ±510 |
| | $GP_{joint}$ | | | | | 138 ±4 |
| Tamil | $GP_{base}$ | 0.565 | 0.641 | 1.52 | 0.72 | 106 ±3 |
| | $GP_{repeat}$ | 0.569 | 0.634 | 1.19 | 0.54 | 2777 ±350 |
| | $GP_{joint}$ | | | | | 107 ±3 |

Table 2 compares GPs that use either only the mean combined reference score or the separate scores from the multiple raters in the training set. The PCC and MSE results may suggest that the proposed $GP_{joint}$ approach yields performance gains over $GP_{base}$, but these may not be significant, with $\rho_{PCC} = 0.065$ and $\rho_{MSE} = 0.036$ for speechocean762, and $\rho_{PCC} = 0.624$ and $\rho_{MSE} = 0.585$ for Tamil. The PCC and MSE compare the prediction against the mean combined reference, thereby assuming that this mean combined reference is correct. Furthermore, this omits consideration of how well the output uncertainty expressed by the model matches the uncertainty between the human raters. The KL divergence is used to assess this uncertainty matching. As a baseline, the discrete KL divergence of the $NN_{categorical}$ model is 1.26 for speechocean762 and 0.70 for Tamil. The results show that using the separate scores from multiple training set raters improves both the continuous and discrete KL divergences of the GP, with significances of $\rho_{discrete-KL} < 0.001$ for the discrete KL divergence of both datasets. This suggests that the proposed extension allows the GP predictive density function or distribution to better match that represented by the collection of scores from the multiple raters. Reiterating from section 5.1, the primary difference between the standard and proposed approaches is in the estimation of the hyper-parameters during training.

Multiple output samples in the training set can also be considered by repeating the inputs in $GP_{repeat}$, as is described in section 4.1. The proposed $GP_{joint}$ approach has an advantage of a reduced number of operations required to perform matrix inversion. This computational saving was assessed by repeating inference 10 times on a 48 core Intel Xeon Platinum 8268 2.90GHz CPU using the Numpy pseudo-inverse implementation, and the mean and standard deviation of the total time to infer the whole test set accumulated over all threads are shown in the right-side column in table 2. Inference using a GP can be run efficiently by pre-computing the matrix inverse before any test data points are seen. It therefore seems more interpretable to view the total time across the whole test set, rather than the time per data point or real time factor, assuming that the computation time is dominated by matrix inversion. The results show that $GP_{joint}$ is indeed faster to infer from than $GP_{repeat}$. $GP_{joint}$ and $GP_{base}$ require comparable durations to infer from. Although not shown here, this computational saving also benefits the matrix inversion computation during training.

## 7 CONCLUSION

This work has proposed an extension to the GP, to allow for multiple output samples for each input in the training set. The hyper-parameters of such a model can be optimised using a criterion that may be related to a KL divergence. Inference is performed in a fairly similar way to a standard GP, with the main difference being in the hyper-parameters that are optimised differently. This allows the predictive density to better match the uncertainty expressed by the collection of multiple human annotations, and reduces the computational cost compared against a naive approach of repeating the inputs.

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

## A  APPENDIX: RELATION OF CRITERION TO KULLBACK-LEIBLER DIVERGENCE

Maximising the marginal log-likelihood in equation 19 can be related to minimising the Kullback-Leibler (KL) divergence between the marginal likelihood and the reference density function over output samples. To see this relation, consider only a single training data point, $i$, and ignore the correlations between multiple training data points. The marginal log-likelihood for the multiple output references of this single training data point is

$$\mathcal{F} = \log p\left(y_{i1}^{\text{ref}}, \cdots, y_{iR}^{\text{ref}} \middle| \boldsymbol{x}_i\right). \tag{28}$$

If it is assumed that the multiple outputs are independently sampled from $p\left(y_i \middle| \boldsymbol{x}_i\right)$, then equation 28 can be expressed as

$$\mathcal{F} \approx \sum_{r=1}^{R} \log p\left(\mathring{y}_i = y_{ir}^{\text{ref}} \middle| \boldsymbol{x}_i\right) \tag{29}$$

$$\propto \int p^{\text{ref}}\left(\mathring{y}_i = y_i \middle| \boldsymbol{x}_i\right) \log p\left(\mathring{y}_i = y_i \middle| \boldsymbol{x}_i\right) dy_i, \tag{30}$$

where the reference density function is

$$p^{\text{ref}}\left(\mathring{y}_i = y_i \middle| \boldsymbol{x}_i\right) = \frac{1}{R} \sum_{r=1}^{R} \delta\left(y_i, y_{ir}^{\text{ref}}\right). \tag{31}$$

The Dirac delta function is defined as

$$\delta\left(a, b\right) = \begin{cases} \infty & \text{, if } a = b \\ 0 & \text{, otherwise} \end{cases}. \tag{32}$$

The form of equation 30 is of an unnormalised KL divergence. Thus, maximising equation 19 may be related to minimising a KL divergence between $p\left(\boldsymbol{Y}^{\text{ref}} \middle| \boldsymbol{X}\right)$ and the reference density function, under several assumptions.

## B  APPENDIX: STANDARD EVALUATION MEASURES

The Pearson's correlation coefficient is computed as

$$\text{PCC}\left(\widehat{\boldsymbol{y}}^{\star}, \boldsymbol{y}^{\text{ref}}\right) = \frac{\sum_{i=1}^{M}\left(\widehat{y}_i^{\star} - \mathbb{E}\left\{\widehat{\boldsymbol{y}}^{\star}\right\}\right)\left(\widehat{y}_i^{\text{ref}} - \mathbb{E}\left\{\boldsymbol{y}^{\text{ref}}\right\}\right)}{\sqrt{\left[\sum_{i=1}^{M}\left(\widehat{y}_i^{\star} - \mathbb{E}\left\{\widehat{\boldsymbol{y}}^{\star}\right\}\right)^2\right]\left[\sum_{i=1}^{M}\left(\widehat{y}_i^{\text{ref}} - \mathbb{E}\left\{\boldsymbol{y}^{\text{ref}}\right\}\right)^2\right]}}, \tag{33}$$

where the empirical mean is

$$\mathbb{E}\left\{\boldsymbol{y}\right\} = \frac{1}{M}\sum_{i=1}^{M} y_i. \tag{34}$$

The mean squared error is computed as

$$\mathrm{MSE}\left(\widehat{\boldsymbol{y}}^{\star}, \boldsymbol{y}^{\mathrm{ref}}\right) = \frac{1}{M}\sum_{i=1}^{M}\left(\widehat{y}_i^{\star} - y_i^{\mathrm{ref}}\right)^2. \tag{35}$$

## C  APPENDIX: EVALUATING UNCERTAINTY PREDICTION

The GP computes a continuous predictive density at its output. This can be compared to a reference density using an unnormalised continuous KL divergence,

$$\mathrm{KL}_{\mathrm{continuous}} = -\frac{1}{M}\sum_{i=1}^{M}\int p^{\mathrm{ref}}\left(\widehat{y}_i|\widehat{\boldsymbol{x}}_i\right)\log p\left(\widehat{y}_i|\widehat{\boldsymbol{x}}_i, \boldsymbol{y}, \boldsymbol{X}\right) d\widehat{y}_i, \tag{36}$$

where $M$ is the test set size and the reference density is defined in equation 31. A lower value indicates a better agreement. However, the measure is not lower-bounded by zero.

To allow comparison of this agreement with that of a categorical NN model, the GP predictive density can be first discretised by accumulating the likelihoods of scores that would have been rounded to each integer into a discrete posterior,

$$P\left(\acute{y}_i = c\Big|\widehat{\boldsymbol{x}}_i, \boldsymbol{y}, \boldsymbol{X}\right) = \frac{\int_{c-0.5}^{c+0.5} p\left(\widehat{y}_i|\widehat{\boldsymbol{x}}_i, \boldsymbol{y}, \boldsymbol{X}\right) d\widehat{y}_i}{\sum_{c'}\int_{c'-0.5}^{c'+0.5} p\left(\widehat{y}_i'|\widehat{\boldsymbol{x}}_i, \boldsymbol{y}, \boldsymbol{X}\right) d\widehat{y}_i'}, \tag{37}$$

where $c$ are the possible integer score classes, and $\acute{y}$ represents a discrete random variable. The discrete KL divergence between this predictive posterior and the reference distribution can then be computed as

$$\mathrm{KL}_{\mathrm{discrete}} = \frac{1}{M}\sum_{i=1}^{M}\sum_{c} P^{\mathrm{ref}}\left(\acute{y}_i = c|\widehat{\boldsymbol{x}}_i\right)\log\frac{P^{\mathrm{ref}}\left(\acute{y}_i = c|\widehat{\boldsymbol{x}}_i\right)}{P\left(\acute{y}_i = c|\widehat{\boldsymbol{x}}_i, \boldsymbol{y}, \boldsymbol{X}\right)}. \tag{38}$$

The discrete reference distribution can be computed from the multiple human rater scores as

$$P^{\mathrm{ref}}\left(\acute{y}_i = c|\widehat{\boldsymbol{x}}_i\right) = \frac{1}{R}\sum_{r=1}^{R}\partial\left(c, y_{ir}^{\mathrm{ref}}\right), \tag{39}$$

where the Kronecker delta function is

$$\partial\left(a, b\right) = \begin{cases} 1 & \text{, if } a = b \\ 0 & \text{, otherwise} \end{cases}. \tag{40}$$

This assumes that the scores from the multiple raters are provided as integers. The discrete KL divergence is lower-bounded by zero. An analogous discrete KL divergence can be computed when comparing the predictive posterior from a categorical NN against the reference distribution. One of the few available public SLA datasets is speechocean762 (Zhang et al., 2021b). In this dataset's publication of Zhang et al. (2021b), the standardised method of evaluating a model is established as first rounding both the predicted and reference scores to integers, before computing the PCC and MSE. This shifts SLA toward a classification-like task, and further from being a fully regression task. The discretisation into the predictive posterior in equation 37 matches well with this practice.

## D  APPENDIX: SPEECH RECOGNITION MODELS

The speech recognition models used to compute the forced alignment between the audio and the transcripts for both datasets were phonetic hybrid (Bourlard & Morgan, 1994) models trained using the Kaldi toolkit (Povey et al., 2011).

The speech recognition model used to align speechocean762 was constructed following the steps that are used in Zhang et al. (2021b). This trained a model on the Librispeech 960 hours training set (Panayotov et al., 2015), following the standard Kaldi recipe. In this recipe, a 5 layer Time Delay Neural Network (TDNN) (Waibel et al., 1989) acoustic model with Rectified Linear Unit (ReLU) (Glorot et al., 2011) activations, and a total left context of 13 frames and right context of 9 frames, was trained toward the cross-entropy criterion. This had about 19.1 million parameters. The cross-entropy targets were obtained from a forced alignment using a triphone Gaussian mixture model hidden Markov model (Juang, 1985). The input features to the TDNN were 40 dimensional Mel frequency cepstral coefficients (Mermelstein, 1976) and 100 dimensional i-vectors (Dehak et al., 2011). The acoustic model outputs were 5672 tied triphone states (Young et al., 1994).

The Tamil language speech recognition model was trained on 220 hours of closed-talking speech from 710 adults. The acoustic model comprised 3 long short-term memory layers and 6 TDNN layers with ReLU activations, interleaved together (Peddinti et al., 2018), with a total right context of 20 frames. This used 29 dimensional filterbank features, was trained using the lattice-free implementation of the maximum mutual information criterion (Povey et al., 2016), and had around 8.5 million parameters. The outputs were 1888 tied left diphone states.

