# OpenReview forum: "Multiple output samples for each input in a single-output Gaussian process"
_ICLR.cc/2023/Conference — Submitted to ICLR 2023_

### Official Review · Reviewer_PFR9 · 2022-10-17

**Confidence:** 4
**Correctness:** 3
**Technical Novelty And Significance:** 2
**Empirical Novelty And Significance:** 2
**Recommendation:** 3

**Clarity, Quality, Novelty And Reproducibility:**

Reasonably clear. But the main problem formulation could be more directly drawn.  The current draft assumes that Gaussian Processes are optimally useful for the task of confidence estimation.

Quality -- good writing quality, however, a broader evaluation of the approach would substantially strengthen the paper.

Sufficiently detailed to be reproducible


**Strength And Weaknesses:**

Strength
* Clear mathematical extension of the standard GP formulation to train on multiple outputs for a single input.

Weakness
* Unclear that this improvement is solving a problem.
The speech intelligibility task like many subjective evaluations has substantial uncertainty (noise) in the ratings.   It's unclear what a better measure of uncertainty is important to this task in order to understand the necessity of this improvement

* Limited evaluation.
There are a plethora of approaches to measure, predict and account for noise in subjective ratings. In Section 6.5 it is not clear how the KL divergence without multiple input samples is calculated.  (Repeated samples from the vanilla GP?)

How would this approach compare to a naive approach using a fixed covariance across the full data set? Or an only slightly more complicated approach, predicting the variance for each utterance?

The proposed approach still has a Gaussian predictive density function.  This limits the distributions that can be represented by this technique.  How would this compare to a variant whose predictive density function were a mixture of Gaussians similar to "Gaussian Mixture Modeling with Gaussian Process Latent Variable Models" (https://arxiv.org/abs/1006.3640)?

* Efficiency Discussion
Gaussian Processes are computationally expensive.  Are there other approaches to density estimation that are less expensive that can be compared to the proposed? Or that the use of Gaussian Processes can be shown to be more computationally expressive?

**Summary Of The Paper:**

This paper describes an extension to Gaussian Processes to perform training and inference over multiple outputs for each input. The value of this is demonstrated by showing that the extension is a better predictor of the KL divergence of human raters on a speech intelligibility task.

**Summary Of The Review:**

The paper proposes an extension to Gaussian processes to enable training on multiple output samples (while avoiding a naive formulation which is numerically unstable).  While a potentially interesting extension to Gaussian Processes, the paper does not sufficiently demonstrate that this approach is substantially better than other approaches to confidence estimation approaches, or is broadly applicable by evaluating on a single task (speech intelligibility).

---

### Official Review · Reviewer_15fK · 2022-10-21

**Confidence:** 4
**Correctness:** 4
**Technical Novelty And Significance:** 2
**Empirical Novelty And Significance:** 1
**Recommendation:** 3

**Clarity, Quality, Novelty And Reproducibility:**

   It is difficult to follow the paper since several definitions are not introduced. See weaknesses above. Furthermore, the proposed method is fairly simple, but it is explained in a complicated way. It just consists in adding extra likelihood factors, one per each extra observation. Since these factors are Gaussian, the posterior is also Gaussian. Gaussian process inference follows the standard approach.


**Strength And Weaknesses:**

Strengths:

        - The proposed approach may have the advantage of reducing training cost when compared to considering an extended Gaussian process prior which may lead to a non-invertible covariance matrix. However, this is not evaluated.

Weaknesses:

        - The paper has a sloppy notation in which several terms are introduced without clarification. It not clear what Y_ref or y^º_i are. This makes difficult following the paper.

        - The experimental section is weak. Only two datasets are considered and the baseline the authors compared with is very simple.

        - The results show no particular benefit with respect to the simple baseline the authors compare with.



**Summary Of The Paper:**

This paper describes a method for considering multiple outputs associated to a single input point in the context of Gaussian process regression. The paper claims that simply incorporating new observations leads to a singular covariance matrix which cannot be inverted, preventing standard Gaussian process inference. The authors proposed a method that is based on using the standard Gaussian process prior and extra likelihood factors, for the multiple observations. The result is a Gaussian posterior and a Gaussian predictive distribution. The proposed approach is evaluated on two problems. It is compared to a baseline that consists in averaging output observations.


**Summary Of The Review:**

 Overall I think that this is paper presents an idea that is way to simple. It also mentions the problem of inverting the covariance matrix in the naive approach in which the prior is extended also for the extra observations. It is well known that this problem can be simply solved by adding some jitter to the diagonal of the covariance matrix. Therefore, the only advantage of the proposed approach is the computational savings from reducing the covariance matrix of the prior. However, this is not discussed by the authors. Also the experimental comparison is too weak since only two datasets are considered and the benefits with respect to averaging observations is marginal.

---

### Official Review · Reviewer_yWAJ · 2022-10-23

**Confidence:** 3
**Correctness:** 3
**Technical Novelty And Significance:** 2
**Empirical Novelty And Significance:** 2
**Recommendation:** 3

**Clarity, Quality, Novelty And Reproducibility:**

- In the Introduction, it would be better to explain more carefully the importance of the problem setting and the novelty of the proposed method.
- I think it would be better to move all the basic formulas for Gaussian processes (Eq. (8), Eq. (27)-(28), etc.) to the Appendix and rewrite the manuscript focusing on the parts that represent the novelty of the proposed method.
- The proposed model is incremental.

**Strength And Weaknesses:**

S1. The authors are trying to solve a spoken language assessment task using a Gaussian process. The problem addressed herein is important.

W1. The manuscript is not well organized and the technical contributions are somewhat unclear.

W2. The proposed model seems incremental. The proposed model attempts to represent multiple outputs by independent sampling from a single GP. This does not seem to contain any particular novelty.

W3. I think it would be better to explain the importance of problem settings more carefully, for example, by drawing an easy-to-understand diagram.

**Summary Of The Paper:**

This paper proposes an extension of GP that is applicable to situations where a single task is assigned several different output labels. The proposed model is evaluated using real-world datasets.

**Summary Of The Review:**

This paper challenges an interesting task, but lacks novelty. The organization of the paper should also be re-considered.

---

### Official Review · Reviewer_ssqv · 2022-10-24

**Confidence:** 3
**Correctness:** 2
**Technical Novelty And Significance:** 2
**Empirical Novelty And Significance:** Not applicable
**Recommendation:** 1

**Clarity, Quality, Novelty And Reproducibility:**

Clarity: The descriptions are clear.
Quality: If my concern written in weakness is true, technical quality is weak.
Novelty: Same as quality.
Reproducibility: OK.


**Strength And Weaknesses:**

Strength:

The paper is well-organized and easy to follow.

Weaknesses:

To be honest, I do not understand why the proposed approach is required. Although the author mentioned that the stacked kernel approach does not work because eq(13) is not full rank. However, as far as the noise variance is nonzero \sigma > 0, the predictive distribution of GPR can be calculated even when (13) is not full rank (inverse is required only for (K + sigma^2 I), not for K). Obviously, it is usual that, when \sigma > 0, multiple different observations can be obtained for the identical x. The standard GPR framework can deal with this situation.

**Summary Of The Paper:**

The paper proposes an extension of Gaussian process regression (GPR) for the case that multiple output values can be obtained for one common input. Multiple output values follow the standard GPR noise model, and the authors consider KL based criterion for hyper-parameter optimization.

**Summary Of The Review:**

The paper is well-organized, but I currently do not find the fundamental motivation for introducing the proposed method. The standard GPR framework can handle the problem setting that the authors seemingly claim it cannot.

---

### Decision · Program_Chairs · 2023-01-20

**Decision:**

Reject

**Justification For Why Not Higher Score:**

* novelty is lacking

**Justification For Why Not Lower Score:**

* topic is of interest

**Metareview: Summary, Strengths And Weaknesses:**

all authors agree this paper should be rejected. Here is the summary.

Strengths:
* well organized
* problem is important

Weakness:
* novelty is lacking
* explanations are unclear and lacking
* unclear the problem really being solved.